



# Measurement report: Methane (CH$_4$) sources in Krakow, Poland: insights from isotope analysis

Malika Menoud[1], Carina van der Veen[1], Jaroslaw Necki[2], Jakub Bartyzel[2], Barbara Szénási[3], Mila Stanisavljević[2], Isabelle Pison[3], Philippe Bousquet[3], and Thomas Röckmann[1]

[1]Institute for Marine and Atmospheric research Utrecht (IMAU), Utrecht University, Utrecht, The Netherlands
[2]Faculty of Physics and Applied Computer Science, AGH University of Science and Technology, Kraków, Poland
[3]Laboratoire des sciences du climat et de l'environnement (LSCE), Université Paris-Saclay, CEA, CNRS, UVSQ, Gif-sur-Yvette, France

**Correspondence:** Malika Menoud (m.menoud@uu.nl)

**Abstract.** Methane (CH$_4$) emissions from human activities are a threat to the resilience of our current climate system, and to the adherence of the Paris Agreement goals. The stable isotopic composition of methane ($\delta^{13}$C and $\delta^2$H) allows to distinguish between the different CH$_4$ origins. A significant part of the European CH$_4$ emissions, 3.6 % in 2018, comes from coal extraction in Poland; the Upper Silesian Coal Basin (USCB) being the main hotspot.

Measurements of CH$_4$ mole fraction ($\chi$(CH$_4$)), $\delta^{13}$C and $\delta^2$H in CH$_4$ in ambient air were performed continuously during 6 months in 2018 and 2019 at Krakow, Poland, 50 km east of the USCB. In addition, air samples were collected during parallel mobile campaigns, from multiple CH$_4$ sources in the footprint area of the continuous measurements. The resulting isotopic signatures from sampled plumes allowed us to distinguish between natural gas leaks, coal mine fugitive emissions, landfill and sewage, and ruminants. The use of $\delta^2$H in CH$_4$ is crucial to distinguish the fossil fuel emissions in the case of

Krakow, because their relatively depleted $\delta^{13}$C values overlap with the ones of microbial sources. The observed $\chi$(CH$_4$) time series showed regular daily night-time accumulations, sometimes combined with irregular pollution events during the day. The isotopic signatures of each peak were obtained using the Keeling plot method, and generally fall in the range of thermogenic CH$_4$ formation - with $\delta^{13}$C between -55.3 and -39.4 ‰ V-PDB, and $\delta^2$H between -285 and -124 ‰ V-SMOW. They compare well with the signatures measured for gas leaks in Krakow and USCB mines.

The CHIMERE transport model was used to compute the CH$_4$ and isotopic composition time series in Krakow, based on two emission inventories. The $\chi$(CH$_4$) are generally under-estimated in the model. The simulated isotopic source signatures, obtained with Keeling plots on each simulated peak using the EDGAR v5.0 inventory, indicate that a higher contribution from fuel combustion sources in EDGAR would lead to a better agreement. The isotopic mismatches between model and observations are mainly caused by uncertainties in the assigned isotopic signatures for each source category, and the way they

are classified in the inventory. These uncertainties are larger for emissions close to the study site, which are more heterogenous than the ones advected from the USCB coal mines. Our isotope approach proves to be very sensitive in this region, thus helping to evaluate emission estimates.



## 1 Introduction

The emissions of greenhouse gases from human activities are the main cause of the current warming of our Earth's climate. It is
urgent to decrease these emissions in order to minimise the negative consequences (IPCC (2018)). The second most important
anthropogenic greenhouse gas after carbon dioxide ($CO_2$) is methane ($CH_4$; IPCC (2018)). $CH_4$ has a Global Warming Potential (GWP; integrated radiative forcing relative to that of $CO_2$ per kg of emission) of 86 over a 20 year time horizon, including
carbon cycle feedbacks (IPCC (2013)). On a global scale, 23 % of the additional radiative forcing since 1750 is attributed to
$CH_4$, whereas total $CH_4$ anthropogenic emissions represent only 3 % of the ones of $CO_2$ in term of carbon mass flux (Etminan
et al. (2016)). In recent years, the total $CH_4$ emissions have been rising: they increased by 5 % in the period 2008-2017 (and 9
% in 2017), compared to the period 2000-2006 (Saunois et al. (2020)). It is not clear which sources have caused these changes,
but Saunois et al. (2020) estimated anthropogenic emissions to represent 60 % of the total emissions of the past 10 years. An
effective reduction of $CH_4$ emissions requires knowledge of the locations and magnitudes of the different sources.

Atmospheric measurements of greenhouse gases at several locations have been used to investigate the rates, origins, and
variations in emissions. However, for methane, these are not always in agreement with what is reported in the emissions
inventories (Saunois et al. (2020)). Isotopic measurements are used to better constrain the sources of methane at regional
(e.g. Levin et al. (1993), Tarasova et al. (2006), Beck et al. (2012), Röckmann et al. (2016), Townsend-Small et al. (2016),
Hoheisel et al. (2019), Menoud et al. (2020b)) and global (e.g. Monteil et al. (2011), Rigby et al. (2012), Schwietzke et al.
(2016), Schaefer et al. (2016), Nisbet et al. (2016), Worden et al. (2017), Turner et al. (2019)) scales. Indeed, the different
$CH_4$ generation pathways lead to different isotopic signatures (Milkov and Etiope (2018), Sherwood et al. (2017), Quay et al.
(1999)). Recently, instruments for continuous measurements of the isotopic composition of $CH_4$ have been developed (Eyer
et al. (2016), Chen et al. (2016), Röckmann et al. (2016)) and used to characterise the main sources of a specific region
(Röckmann et al. (2016), Yacovitch et al. (2020), Menoud et al. (2020b)). Using model simulations, the observations can be
used to evaluate the partitioning of the different sources reported in the inventories (Rigby et al. (2012), Szénási (2020)).

Saunois et al. (2020) stated the need for more measurements in regions where very few observations are available so far. In
Europe, inventories report high $CH_4$ emissions from Poland (European Environment Agency (2019)). In 2018, they represented
10 % of total European Union emissions, with more than 48 Mt $CO_2$ eq.. Half of these are from the energy sector, among which
% are due to the exploitation of underground coal mines (National Centre for Emission Management (KOBiZe) and Institute
of Environmental Protection - National Research Institute (2020), Swolkień (2020)). The Upper Silesian Coal Basin (USCB),
where most mining activity occurs in Poland, is certainly a $CH_4$ emission hotspot in Europe. Atmospheric measurements at the
USCB were mostly performed in the recent years (Swolkień (2020), Luther et al. (2019), Gałkowski et al. (2020), Fiehn et al.
(2020)), and focused on the coal extraction activities. The area covered by the USCB includes other sources of methane, such
as ruminant farming and waste degradation. In this study we investigate whether we can use isotopic signals to distinguish the
different sources. We attempted to detect them from Krakow, where we wanted to establish the main $CH_4$ sources affecting
such a densely populated area. Finally, we investigate whether we can use this tool to put constrains on the emission inventories
in order to improve them.





To this end, we carried out and investigated quasi-continuous measurement of $CH_4$ mole fraction, $^{13}C/^{12}C$ and $^2H/^1H$ isotopic ratios of $CH_4$ in ambient air during 6 months at a fixed location in Krakow, Poland. Time series of these isotopic ratios were also simulated with an atmospheric transport model, based on two different emission inventories. The local $CH_4$ sources were sampled during several mobile measurement campaigns, to determine their isotopic signatures and compared with the ambient measurements.

## 2 Methods

### 2.1 Target region and time period

The region of study is characterised by the presence of a large coal mining region: the Upper Silesian Coal Basin (USCB). It gathers 20 active coal mines spread over an area of 1100 km$^2$ (Swolkień (2020)), and is located about 50 km west of Krakow (Fig. 1). Other potential $CH_4$ sources around Krakow are from waste management and wastewater treatment facilities, industrial activity, energy production and the natural gas distribution network. Large-scale agriculture activities are not characteristic for this area, and only very few cattle farms could be located.

Ambient air measurements were performed from the Faculty of Physics and Applied Computer Science building, at AGH university in Krakow (50°04'01.1"N, 19°54'46.9"E, Fig. 1). We used a 1/2" o.d. Synflex Dekabon air intake line that draws air from the top of a mast on top of the building (35 m above ground level, 255 m a.s.l.) down to the laboratory of the Environmental Physics Group. A fraction of the incoming air was directed via a T-split to the IRMS system in the period from September 14$^{th}$, 2018 to March 14$^{th}$, 2019.

Individual emission locations of methane were visited in and around the city of Krakow, and in the USCB during mobile surveys. The surveys were performed in May 2018 (from 24$^{th}$ to 29$^{th}$), February 2019 (from 5$^{th}$ to 7$^{th}$) and March 2019 (from 20$^{th}$ to 22$^{th}$). We visited the following areas, which are shown on the map in Fig. 1: the Silesian coal basin, Barycz landfill, the industrial park, the city center and other residential areas, and rural areas west of the city.

### 2.2 Sampling

The mobile surveys were conducted with an Integrated Cavity Output Spectroscopy (ICOS) instrument (MGGA - 918, Microportable Greenhouse Gas Analyser, Los Gatos Research, ABB) onboard of a car. An 1/8" Parflex inlet line was placed on top of the vehicle's roof and connected to the analyser. Real time $CH_4$ mole fractions were read on a tablet screen, so that an emission plume could be detected while driving. If the increase was higher than 200 ppb above background, we drove back to the plume and took one to three samples directly from the outflow of the $CH_4$ analyser, using sampling bags (Supel™-Inert Multi-Layer Foil, Sigma-Aldrich Co. LLC).

One or two samples were taken where we observed the lowest $\chi(CH_4)$ during each survey day, in order to obtain the background we can associate with the plumes sampled each day in a certain area.



The samples collected during the mobile surveys were analysed on the same IRMS instrument as the ambient air, partly when it was installed in Krakow, and partly when it was installed back at the IMAU lab in Utrecht.

### 2.3 Isotopic measurements

The $^{13}$C/$^{12}$C and $^2$H/$^1$H isotope ratios in $CH_4$ are expressed as $\delta^{13}$C and $\delta^2$H (deuterium), respectively, in per mil (‰), relative to the international reference materials, Vienna Pee Dee Belmnite (V-PDB) for $\delta^{13}$C and Vienna Standard Mean Ocean Water (V-SMOW) for $\delta^2$H.

The isotopic composition measurements were performed using an Isotope Ratio Mass Spectrometry (IRMS) system, as the one described in Röckmann et al. (2016) and Menoud et al. (2020b). Ambient air or sample air measurements were interspersed

with measurements of a reference cylinder filled with air with assigned composition of $\chi(CH_4)$ = 1950.3 ppb, $\delta^{13}$C-$CH_4$ = -47.82 $\pm$ 0.09 ‰ V-PDB, and $\delta^2$H-$CH_4$ = -92.2 $\pm$ 1.8 ‰ V-SMOW. The reference air bottle was previously calibrated against a reference gas measured at the Max Planck Institute in Jena, Germany (Sperlich et al. (2016)).

The extraction and measurement steps are illustrated in Fig. S1 of the supplementary material. Each measurement of either $\delta^{13}$C or $\delta^2$H returned a value of $CH_4$ mole fraction ($\chi(CH_4)$). A $\delta^{13}$C-$CH_4$ or $\delta^2$H-$CH_4$ value in ambient air was obtained

on average every 27 minutes during the periods of normal operation. In addition to unexpected disturbances or failures, the scheduled replacement of several components (oven catalysts, chemical dryer, fittings, etc.) and the regular flushing and heating of the traps required to stop the measurements for a few hours up to a few days, several times during the study period.

The air was simultaneously measured by a CRDS instrument (G2201-i Isotopic Analyzer, Picarro) installed in the same lab as the IRMS system and drawing air from the same inlet tube. Time series of $CH_4$ mole fractions from both instruments were

compared for quality control.

### 2.4 Meteorological data

Data on the hourly wind direction, speed, and temperature were obtained from an automatic weather station (Vaisala WXT520, Vaisala inc.) installed on the same building as the inlet line (220 m a.s.l.). The station is operated by the Environmental Physics Group, and the data is publicly available at http://meteo.ftj.agh.edu.pl/archivalCharts (registration required). Data on PM10

concentrations is also available on the same platform at this location.

### 2.5 Modelling

Time series of $\delta^{13}$C and $\delta^2$H -$CH_4$ were generated from simulated $CH_4$ mole fractions using the CHIMERE atmospheric transport model (Menut et al. (2013), Mailler et al. (2017)), driven by the PYVAR system (Fortems-Cheiney et al. (2019)). CHIMERE is a three-dimensional Eulerian limited-area chemistry-transport model for the simulation of regional atmospheric

concentrations of gas-phase and aerosol species.

The simulations were carried out at a horizontal resolution of 0.1 ° x 0.1 ° in a domain covering Poland and nearby countries; [46.0° - 55.9°] in latitude and [12.0° - 25.9°] in longitude. The meteorological data used to drive CHIMERE were obtained





from the European Centre for Medium-Range Weather Forecast (ECMWF) operational forecast product. The boundary and initial concentrations of $\chi(\mathrm{CH_4})$ were taken from the analysis and forecasting system developed in the Monitoring Atmospheric

Composition and Climate (MACC) project (Marécal, 2015). They were used to derive the background mole fractions.

The $\mathrm{CH_4}$ emission rates over the domain are reported in emission inventories, following a bottom-up approach. We used two anthropogenic emission inventories for this study: EDGAR v5.0 (Emission Database for Global Atmospheric Research, Crippa et al. (2019)) and CAMS-REG-GHG v4.2 (The Copernicus Atmosphere Monitoring Service REGional inventory for Air Pollutants and GreenHouse Gases, Granier et al. (2012)). We classified the emissions in 6 anthropogenic source categories

based on the European Environment Agency (EEA) greenhouse gas inventory common reporting format (CRF, European Environment Agency (2019)). We considered one additional category for natural wetland emissions, which are obtained from the ORCHIDEE-WET process model (Ringeval et al. (2011)). The classifications used in CHIMERE and the corresponding categories in the inventories are summarised in Table 1.

The isotopic values at each time $t$ were calculated using the following formula:

$$\delta_t = \frac{1}{c_t} \sum_i^{n_S} (c_{S,i} * \delta_{S,i})$$

with $c_t$ the total mole fraction from the model at time $t$, $c_S$ the modelled mole fraction attributed to the source $S$, and $\delta_S$ the source signature of each specific source $S$. In this mass balance, the contribution of the background is treated as a source with assigned isotopic composition. All the assigned source signatures are defined in Table 1.

## 2.6 Isotopic signatures assigned to $\mathrm{CH_4}$ elevations in the long-term time series

Periods of methane enhancement were identified from the $\chi(\mathrm{CH_4})$ time series using a peak extraction method, based on the detection of local maxima from comparison with the neighbouring points. The peaks were selected based on two criteria:

- the peak has a minimal amplitude of 100 ppb

- the peak is composed of at least three data points, from the maximum to a relative height of 0.6 times the peak height.

In order to define the background more robustly, we included additional data from the 10[th] lower percentile of $\chi(\mathrm{CH_4})$ in a

window of $\pm$ 24 h around the maximum of each peak. The Keeling plot method was thus applied to the data points in the peak, together with the neighbouring background data.

The Keeling plot is a mass balance approach (Keeling (1961), Pataki et al. (2003)), considering the measured $\mathrm{CH_4}$ ($m$) in ambient air as the sum of a contribution of $\mathrm{CH_4}$ from an emission source ($s$) and a background ($bg$) $\mathrm{CH_4}$, such that:

$$c_m = c_{bg} + c_s$$
$$c_m \delta_m = c_{bg} \delta_{bg} + c_s \delta_s$$

with $c$ and $\delta$ referring to the mole fraction and isotopic signatures of either $^{13}\mathrm{C}$ or $^2\mathrm{H}$, respectively. Re-arranging the formula leads to:

$$\delta_m = c_{bg} * (\delta_{bg} - \delta_s)(1/c_m) + \delta_s$$





We assumed the background mole fraction and isotopic composition to be stable over the time period of each peak. In this case, $\delta_s$ is given by the y-intercept of the regression line, when plotting $\delta_m$ against $1/c_m$.

To derive an average source signature for the entire dataset, the Miller-Tans approach was used (Miller and Tans (2003)), because the hypothesis of stable background is violated. This method is based on the following formula:

$$c_m \delta_m = \delta_s c_m - c_{bg}(\delta_{bg} - \delta_s)$$

where $\delta_s$ is now given by the slope of the regression line, when plotting $c_m * \delta_m$ against $c_m$.

The linear regressions were made with the Bivariate Correlated Errors and intrinsic Scatter (BCES) fitting method (Akritas and Bershady (1996)), to allow for measurement errors in both variables. An isotopic signature was obtained for each regression. The corresponding uncertainty is always given as 1 standard deviation of the estimated parameter (intercept for the Keeling plot or slope for the Miller-Tans plot).

The method was applied to both $\delta^{13}$C and $\delta^2$H measurement results. If two peaks were detected within a 6 hour time window in the $\delta^{13}$C and $\delta^2$H time series, they were considered one single peak and the two signatures were allocated to it. The same method was also used for the modelled $\chi(CH_4)$ time series, to allow the comparison of modelled and measured source signatures.

## 3    Results and discussion

### 3.1    Observed time series

The observed time series are shown in Fig. 2, together with measurements from the KASLAB laboratory at the top of Kasprowy Wierch, a mountain in southern Poland (49°13'57"N, 19°58'55"E, 1989 m a.s.l.; Necki et al. (2013)). We note that in the period February-March 2019, we observed a mismatch of about 80 ppb between the IRMS-derived and simultaneous CRDS $\chi(CH_4)$ measurements in the same laboratory (shaded area in Fig. 2). A mismatch in mole fraction can potentially affect the Keeling

plot intercepts, and we investigated possible artefacts using various attempts for correction. We realised that the effect of these corrections on the isotopic source signatures is small compared to the observed range (average peak $\delta^{13}$C and $\delta^2$H changed by 0.1 %; differences per peak are shown in Fig. S2). As no obvious reason for a malfunction of the IRMS instrument could be detected, we decided to use the original data without correction. The peaks in $\chi(CH_4)$, compared to the background measured at Kasprowy Wierch, reflect pollution events in Krakow or advected to the measurement site. The maximum $\chi(CH_4)$ value

was 3634 ppb, measured on October 19[th], 2018 at 5:30 am. Simultaneous changes are visible in the $\delta^{13}$C and $\delta^2$H time series. Increased $\chi(CH_4)$ were always linked with a lower $\delta^2$H, but for $\delta^{13}$C the measured values could be higher or lower.

The general background threshold is 1986.0 ppb, which corresponds to the 10[th] lower percentile of the entire dataset. We have found that 70.5 % of the background values ($\chi(CH_4) < 1986.0$ ppb) occurred during daytime. The dominant feature in the CH$_4$ time series is indeed the presence of a diurnal cycle: $\chi(CH_4)$ elevations regularly occurred during the night. This is

due to the lowering of the boundary layer when the temperature decreases in the evening. The morning and evening variations in $\chi(CH_4)$ were negatively correlated with the temperature data we obtained at the study site. In addition, there were isolated





pollution events occurring on top of the night-time accumulation. Between the emission peaks, $\chi(CH_4)$ generally went back to a local background level.

The night-time accumulation was particularly visible in the period September 14$^{th}$ to mid-November 2018, and shown in the supplementary material (Fig. S3). Similar nighttime elevations are also visible in the observations of other pollutants such as PM10 at the study location. There was a clear difference in local temperature before and after November 15, 2018: the average air temperature decreased from $12 \pm 5.3$ °C to $2.1 \pm 4.4$ °C and the dew point temperature from $5.3 \pm 3.4$ °C to $-3.9 \pm 3.4$ °C until the end of the measurements. The period before mid-November will be referred to as fall throughout the paper.

The wind directions at the study site were combined with the CH$_4$ measurement data in Fig. 3; and with wind speeds in Fig. S4 of the supplementary material. The spread of the wind directions was similar for most of the months: mainly from the west, and partly from east/north-east. An exception was November 2018, when most of the wind was from the east/north-east direction. March 2019 was characterised by winds from the west only, and at particularly strong speeds (on average 3.1 m/s, compared to 1.8 m/s for the other months; Fig. S4). The average CH$_4$ diurnal cycle, defined as the prominence of night peaks, was on average 334 ppb throughout the entire time period, but only of 195 ppb when the winds were $> 2.5$ m/s. This decrease in amplitude with higher wind speeds was not influenced by the direction of the wind. During fall, 84 % of the peaks were observed at night and associated with low wind speeds, which suggests the influence of local pollution sources, and a relatively low influence of the wind direction.

The average isotopic values of the background were $\delta^{13}C = -47.8 \pm 0.16$ ‰, and $\delta^{2}H = -90.0 \pm 3.0$ ‰. The CH$_4$ elevations were associated with consistently more negative $\delta^{2}H$, but varying $\delta^{13}C$. This indicates that the sources were sometimes higher in $\delta^{13}C$ compared to the ambient CH$_4$ (i.e. $\delta^{13}C > -47.8$ ‰). In contrast, all CH$_4$ elevations were associated with lower $\delta^{2}H$ during the entire time period.

### 3.2 Modelled time series

The CH$_4$ time series obtained with CHIMERE for the grid cell containing the observation site, are shown in Fig. 4. We first compared the CH$_4$ mole fractions measured at Krakow and modelled by CHIMERE in Fig. 5. They show a poor correlation (Person's correlation coefficients $r^2 = 0.527$ and $r^2 = 0.514$, for model calculations using the EDGAR v5.0 and CAMS-REG-GHG v4.2 inventories, respectively; Fig. 5.A). The model globally under-estimates the measured $\chi(CH_4)$ significantly, with a root mean square error (RMSE) of 164.4 ppb and 173.4 ppb for EDGAR and CAMS, respectively. Yet we see that modelled $\chi(CH_4)$ can sometimes be larger than the observations, which is usually due to a shift in the timing of a pollution event (Fig. 4). The wind data used in the model are generally in good agreement with the wind measurements at the study site, but small discrepancies can partly explain the differences in the timing of the peaks. The time series are best reproduced during the fall 2018, using EDGAR v5.0 ($r^2 = 0.648$; Fig. 5.B). As mentioned in section 3.1, this period shows a more regular pattern of night-time elevations of relatively similar amplitudes compared to the winter period. This is better reproduced by the model (Fig. 4). However, the two highest $\chi(CH_4)$ measurements were observed in this period (October 18, and November 3, 2018) and were not modelled to the same level (points on the lower right, Fig. 5.B). These events largely contribute to the general model under-estimation when only considering the fall data.





In winter, the pollution events were less regular, with a less predictable $\chi(CH_4)$ diurnal cycle. The mismatch in the timing of pollution events caused an over-estimation by the model (points on the upper left, Fig. 5.B). The general slope is still lower than 1, and the fit is worse than during fall. There is a general under-estimation of the $CH_4$ mole fractions at Krakow by the model. This could be explained by the model time series being hourly averages, compared to the observations of sampled

air. To account for this bias, we compared the model data with observations that are also averaged over a 1h window, and/or interpolated to the modelled times. This had no effect on the correlation coefficients, suggesting a minor impact of the temporal representation error. But potential $CH_4$ sources in the close surroundings of the laboratory could affect the measurements compared to the model, where they are diluted over the 11 km grid cell. This spatial representation error could explain $\chi(CH_4)$ under-estimation in CHIMERE. Other potential reasons of misfit include errors in the transport modelling or too low emissions

in the inventories. Szénási (2020) identified the emission inventories as the main source of discrepancies between CHIMERE results and measured time series at two other European locations. The implications on the two inventories are discussed in detail in section 3.4.

Time series of $\delta^{13}C$ and $\delta^2H$ in $CH_4$ show negative or positive excursions relative to the background, and are linked to $\chi(CH_4)$ peaks (Fig. 4). When using CAMS-REG-GHG v4.2, $\delta^{13}C$ and $\delta^2H$ are always negatively correlated with $\chi(CH_4)$. But

when using the EDGAR v5.0 inventory, $\delta^{13}C$ values are closer to the background, and only $\delta^2H$ values are systematically lower at higher $\chi(CH_4)$. The isotopic discrepancies will be analysed in detail in relation to the source partitioning in the inventories, and the signatures we assigned to each source in section 3.4.

### 3.3 Isotopic source signatures

A total of 126 and 156 peaks were identified in the $\delta^{13}C$ and $\delta^2H$ time series, respectively. 114 peaks were measured commonly

by both isotope lines. From the Keeling plot applied to each of the peaks, we obtained the source signatures of the corresponding accumulation events. They can be compared with the determined isotope signatures of the sources sampled in the surrounding area (Fig. 6.A).

### 3.3.1 Isotopic characterisation of the surrounding sources

The results from individual sites are presented in Table 2, and shown in the supplementary material (Fig. S6.A). They are in

good agreement with the ranges defined for the different categories in the literature (Sherwood et al. (2017)). Biogenic sources (a landfill, 3 manholes and a cow barn) correspond to the acetate fermentation pathway, characterised by relatively depleted $\delta^{13}C$ ($< -50$ ‰) and $\delta^2H$ ($< -275$] ‰; Milkov and Etiope (2018)). The landfill $CH_4$ is isotopically more enriched than the cow barn. This can be due to an isotope fractionation from diffusion and oxidation in the soil layers (De Visscher (2004), Bakkaloglu et al. (2021)). The fossil fuel $CH_4$ emissions we sampled were from coal exploitation and use of natural gas. The natural gas

distribution network was sampled outside of compressor stations, close to gas stations and supply valves in residential areas. The results ranged between [-52.4, -44.1] ‰ for $\delta^{13}C$, and [-226, -176] ‰ for $\delta^2H$. To check for temporal variations, two plumes were sampled at an interval of 6 weeks, on February 5 and March 19, 2019. The $\delta^{13}C$ results agreed within $\pm 5$ %, and the $\delta^2H$ within $\pm 10$ %. One sample was directly taken from the gas supply pipe at the AGH lab in March 2019. The pure





gas was 3.4 ‰ and 13 ‰ more depleted in $\delta^{13}$C and $\delta^2$H, respectively, than the average from all leaks (signature in brackets
in Table 2), but still falls in the same range as the sampled leaks. The network gas composition can change in time because the
proportions of gas from several origins varies. Gas migrating in the distribution network can undergo secondary processes such
as oxidation, that influence the isotopic signatures, usually towards more enriched values. Isotopic variations among network
gas leaks were also observed previously in other cities (Zazzeri et al. (2017), Maazallahi et al. (2020), Defratyka et al. (2021)).

CH$_4$ emissions from manholes were often observed in the Krakow urban area. The resulting isotopic signatures do not
indicate one clear origin, and were divided in two groups with distinct $\delta^2$H (Table 2). While the isotopically depleted signatures
observed at 3 locations likely come from the sewage system, with a $\delta^2$H < -250 ‰, the 5 others contain particularly enriched
thermogenic gas ($\delta^{13}$C between [-42.2; -33.3] and $\delta^2$H [-201; -148] ‰; Fig. S6.A). We hypothesise that this indicates leakage
of natural gas from the distribution pipes to the sewage network, which is sometimes further oxidised leading to even more
enriched isotope signatures.

For most emission plumes, we could not visually identify an obvious CH$_4$ source. The isotopic signatures of these "unknown"
sources range from -57.3 to -42.4 ‰ V-PDB for $\delta^{13}$C and from -291.5 to -88.2 ‰ V-SMOW for $\delta^2$H. The $\delta^2$H range is
particularly large, indicating the presence of both fossil fuel and biogenic sources. The average $\delta^2$H is > 200 ‰, suggesting a
major influence from fossil fuel sources. The $\delta^{13}$C is in good agreement with the signature found for natural gas (Table 2 and
Fig. S6.A), and since most of these locations were close to roads and urban settlements, it is likely that they were natural gas
leaks.

The isotope signatures from coal mine ventilation shafts and residential gas leaks sampled in this study fall in the same
range: $\delta^{13}$C between -58.9 and -28.0 ‰ V-PDB, and $\delta^2$H between -254 and -139 ‰ V-SMOW, although coal CH$_4$ has a
wider isotopic range. The values of $\delta^{13}$C < -60 ‰ confirmed the presence of microbial gas in the USCB, and reported in
the literature (Kotarba (2001), Kotarba and Pluta (2009) and Kedzior et al. (2013); Fig. S6.A). Most $\delta^{13}$C values from coal
mines in this study were found between -58 ‰ and -45 ‰, which also indicates a contribution from microbial gas sources,
although in our measurements all $\delta^{13}$C signatures from time series peaks and sampled shafts were > -60 ‰. Some of the
locations sampled in by Kotarba (2001) were re-visited in this study. However, their method used direct sampling of CH$_4$ from
different coal layers, aiming at representing the variety in the origin of the gas reservoirs. Our approach was to sample outside
the shafts, to obtain the isotopic signature of CH$_4$ emissions from these shafts to the atmosphere. The very depleted $\delta^{13}$C
values obtained in these previous studies confirm the presence of purely microbial gas reservoirs in the USCB coal deposits,
but our results show that thermogenic gas represents a larger part of the fugitive emissions from mining activities in this area
than indicated by Kotarba (2001; Fig. 6.A). The heterogeneity of isotopic signatures from coal mining activities in the USCB
reflects the geological complexity of the area. Secondary processes (desorption, diffusion or oxidation) also influence the CH$_4$
isotopic composition, and depend on external parameters such as physical characteristics of the coal reservoirs and the soil
layers (Niemann and Whiticar (2017)). These represent additional difficulties as regards the isotopic characterisation of coal
associated CH$_4$ emissions.

The $\delta^2$H signatures allow us to identify the CH$_4$ emissions from microbial fermentation: values below -250 ‰ are indicative
of the anaerobic fermentation pathway, such as in the rumen of cows or during waste degradation. Except for one shaft with





$\delta^2$H = -254 $\pm$ 0.01 ‰ (possibly very early mature thermogenic gas in deep formations, or a late stage of biodegradation if

close to the surface; Milkov and Etiope (2018)), both literature data and our sampled shafts have a $\delta^2$H > -250 ‰. This is also true for emissions from the natural gas network, confirming their fossil fuel origin. In the USCB region, $\delta^2$H signatures seem to be more suitable than $\delta^{13}$C values for source apportionment, similar to recent studies made in European cities (in Hamburg by Maazallahi et al., 2020, and in Bucharest by Fernandez et al., 2021)

### 3.3.2 Isotopic characterisation of CH$_4$ in ambient air

The isotopic signatures of the CH$_4$ pollution events observed in Krakow during the study period are shown in Fig. 6. $\delta^{13}$C varied between -55.3 and -40.0 ‰ V-PDB, and $\delta^2$H between -267 and -127 ‰ V-SMOW. As mentioned above, the observed $\delta^{13}$C either increased or decreased with higher $\chi$(CH$_4$), indicating source signatures either lower or higher than the background value. Yet $\delta^{13}$C signatures stayed within $\pm$ 8 ‰ from the background, thus never reaching extreme values. The proportion of CH$_4$ peaks enriched in $\delta^{13}$C with respect to the background was 40.5 %. In contrast, the observed $\delta^2$H values were always

more depleted than ambient. The overall source signatures resulting from the Miller-Tans analysis using all the data points were $\delta^{13}$C = -48.3 $\pm$ 0.19 ‰, and $\delta^2$H = -203 $\pm$ 0.95 ‰ (Fig. S5). The comparison with typical signatures of the different CH$_4$ formation processes indicates that most of these events were from thermogenic sources (Fig. S6.B). When compared with isotope signatures of the surrounding sources (Fig. 6.A), the source signatures from the long-term time series match the range of coal mine and natural gas emissions the best. Fig. 6.B shows that most pollution events associated with strong winds fall in

the range of more depleted $\delta^{13}$C signatures. They were also all advected from west of Krakow, where the USCB is located (Fig. 1). In fact, the $\delta^2$H signatures exclude a large contribution from potential biogenic sources, and point towards the emissions from coal mines in Silesia.

In Röckmann et al. (2016) and Menoud et al. (2020b), CH$_4$ mole fractions, $\delta^{13}$C and $\delta^2$H isotopic signatures in ambient air were measured at two locations in the Netherlands. The time series covered 5 months in 2014-2015 and 2016-2017, at

Cabauw and Lutjewad, respectively. The average isotopic signatures were -60.8 $\pm$ 0.2 ‰ and -298 $\pm$ 1 ‰ at Cabauw and -59.5 $\pm$ 0.1 ‰ and -287 $\pm$ 1 ‰, for $\delta^{13}$C and $\delta^2$H, respectively. The main sources contributing to the CH$_4$ emissions in the Netherlands are cattle farming and waste management. These are biogenic sources, with isotopic signatures representative for the microbial fermentation origin. CH$_4$ of fossil fuel origin had a minor contribution there, which contrasts a lot with the results from Krakow. Such drastic differences in the isotopic signals of the same greenhouse gas show how a region-specific analysis

is crucial to effectively constrain atmospheric emissions.

In Fig. 7, the results of CH$_4$ mole fraction, peak source signatures and wind speed and direction are shown in more details for 8 days in November 2018, and 7 days in February 2019, together with model results using EDGAR v5.0.

In general, eastern winds advected CH$_4$ with a relatively enriched $\delta^{13}$C: 60 % were higher than the background $\delta^{13}$C, and all but one were > -50 ‰ V-PDB. In November, the wind was mostly coming from the east (Fig. 3), but elevations were observed

at low wind speed (Fig. 7.A, peaks 4 to 7). These pollution events reflect the general signature of the CH$_4$ emitted in the Krakow urban area and are unlikely to come from coal mines. In Fig. 7.A, the peaks C, D, E and G show a large contribution from the natural gas and from the "other anthropogenic" categories. The latter represents mainly the power generation and transportation





sectors, as well as the manufacture, chemical and metal industries. The main contribution is the energy production from fossil fuels, and we assigned a $\delta^{13}C$ signature corresponding to fossil fuel $CH_4$ to this category (Table 1). The modelled results for these peaks are generally similar to the measured ones. The magnitude of the $\chi(CH_4)$ elevations also matches the observations relatively well: modelled peaks 3, 4, and 5 were 79 ppb, 23 ppb and 14 ppb larger than the observed peaks C, D and E, respectively. Yet for peak C (observed peak 3), the model $\delta^{13}C$ signature is 2.5 ‰ lower than the one from the measurements, and showed a majority of emissions from "other anthropogenic" sources (37 %). Part of these emissions can be from the incomplete combustion of $CH_4$, and such combustion-related emissions have a more enriched $\delta^{13}C$ signature than fossil fuel $CH_4$ (Fig. 6.A). Results from mobile surveys in Paris identified fuel-based residential heating systems as urban $CH_4$ sources, with a slightly more enriched isotopic composition than the local gas leaks (Defratyka et al. (2021)). Therefore, either the proportion of emissions in the "ENB" category, or the $\delta^{13}C$ signature assigned to the "other anthropogenic" emission category were under-estimated. We note that we couldn't characterise this source category by sampling. Uncertainties in the assigned signature are unavoidable when a given category is a combination of different sources; not only the processes have different isotopic signatures, but the contribution from the different sources could change from one pollution event to another. For $\delta^2H$, the agreement between observed and modelled signatures for these November night peaks is good. All fossil fuel and pyrogenic $\delta^2H$ signatures used in this study are relatively close to each other (Table 1), and to the average peak $\delta^2H$ source signature. Thus, the $\delta^2H$ signatures do not allow for a distinction between these two processes.

Some peaks advected at low wind speeds during night are also visible in Fig. 7.B (peaks 9 to 11), and show similarly enriched $\delta^{13}C$ signatures. The wind direction was different for these night peaks between February and November, but the low wind speeds again indicate that this represents the local emission mix. The model time series showed peaks that occurred simultaneously to the measured ones (K and L in Fig. 7.B), although with different $\chi(CH_4)$ maxima than the measurements (-115, -339 and +203 ppb, respectively). For peaks K and L, the source partitioning from the inventory is similar to the other night peaks shown in Fig. 7.A. The $\delta^{13}C$ signatures of these urban emissions are however under-estimated in the model, and so are the $CH_4$ mole fractions, in particular for peak 11 (corresponding to peak L in the model time series). We suggest that at a close distance east of the study site, the share of emissions from the combustion sources is likely under-estimated. These additional emissions could be from residential heating or the energy production sector. The $\delta^2H$ signature of peak 11 (L) also differs significantly between model and measurements. This further indicates that the missing $CH_4$ emissions must be mostly combustion related, because of the relatively enriched $\delta^{13}C$ and $\delta^2H$ we observed (-44.9 ‰ V-PDB and -199 ‰ V-SMOW, respectively, for peak 11).

The $\delta^{13}C$ signatures shifted towards more depleted values after February 19. $\delta^{13}C$ went from -44.9 ± 0.6 ‰ for peak 11 to -50.5 ± 0.7 ‰ for peak 13. Peaks 12 and 13 (respectively M and N in the model), were advected by strong western winds. The share of coal related emissions reported in the inventory increased from peak M compared to peaks K and L, and is supported by the decrease in $\delta^{13}C$ also in the modelled signatures. This confirms a source shift from urban to coal activities further west of Krakow from February 19, 2019. Whenever the EDGAR inventory reported large contributions from coal mine emissions, such as in for peaks F, H, K, M and N (corresponding to 6a, 8, 10a, 12 and 13, respectively), the model wind direction corresponds to the USCB. The associated isotopic signatures were in relatively good agreement for peaks H, M, and N, where coal emissions



represented > 50 % of the total. Small discrepancies ($\pm$ 2 ‰ in $\delta^{13}$C) are explained by the heterogeneity of isotopic signatures from the different mine shafts. This confirms that the average isotopic signatures for this category are well characterised in this

study. For peaks F and K, $\delta^{13}$C values are at least 2 ‰ lower than the observations (peaks 6a and 10a). The share of emissions from the USCB are therefore likely over-estimated in these 2 cases.

Three peaks showed a $\delta^2$H < -260 ‰ V-SMOW, suggesting a larger contribution from biogenic sources (Fig. 6.A). They are associated with large uncertainties, because the peak magnitudes were low. These peaks were not modelled by CHIMERE, using either inventory. They represent isolated pollution events, disconnected from the daily cycle and not particularly related to

a certain wind direction. There could be occasionally larger biogenic emissions such as from a waste facility that are advected to the measurement site. In Fig. 7.B, a depleted $\delta^2$H signature was derived from a small peak (12a). The $\chi(CH_4)$ enhancement was not significant in the time series of $\delta^{13}$C, which suggests a very short pollution event. It still correlated with a short-term change in wind direction towards a more north/north-west origin. Such abrupt changes are not visible in the model wind data, because of its coarser temporal resolution. Based on its clearly biogenic isotopic signal, as well as the wind direction, this event

might reflect the contribution from the 2 large waste treatment facilities located north-west of Krakow (Fig. 1). This needs to be confirmed by observations at higher mole fractions to reduce the uncertainty in the source signature, and be able to derive a signature for $\delta^{13}$C, as we are reaching here our detection limit. Further measurements at this location would be useful to specifically characterise this source.

In addition to the night time accumulations of $CH_4$, we observed occasional $\chi(CH_4)$ peaks during the day, not linked to the

night-time lowering of the boundary layer. $CH_4$ emissions coming from a specific location and advected by strong winds to the measurement site resulted in sharp peaks, such as peak 2 in Fig. 7.A, that are separate from the daily cycle. An increase in wind speed (from 0.7 to 2.2 m/s) and constant wind direction of 251 ° caused a sharp increase in $\chi(CH_4)$ by 1360 ppb, over only 3h. The peak was reproduced by the model (peak A), but with a lower magnitude, which can be explained by the differences in the wind data. The observed source signatures were $\delta^2$H = -190 $\pm$ 5.1 ‰, indicating fossil fuel related emissions, and $\delta^{13}$C =

-50.6 $\pm$ 0.26 ‰, pointing to localised coal mine fugitive emissions. The isotope signatures from the model using the EDGAR inventory differ significantly from the observed ones, even though coal extraction is still indicated as main source. The input source signatures in the model represent all coal related emissions and therefore might fail in reproducing the signature of emissions at the scale of individual sites.

### 3.4 CH$_4$ source partitioning in the inventories linked to isotopic composition

The CH$_4$ emissions for each source category from the inventories over the studied domain and the simulated $CH_4$ mole fractions in the grid-cell of the measurements location are presented in Table 3.

Compared to simulations made with EDGAR v5.0, the modelled isotopic signatures with CAMS-REG-GHG v4.2 show that the CH$_4$ sources are always more isotopically depleted in $\delta^{13}$C (section 3.2, Fig. 4). When looking at the source partitioning between the 2 inventories, this can be explained by the much higher contribution from waste emissions when using the CAMS

inventory (Table 3). These emissions have a particularly large influence at our study site (43.8 % of total added mole fraction), whereas the share in the emissions is not so large over the entire domain (26.2 % of total emissions). The emissions maps of





both inventories are shown in Fig. S7 of the supplementary material. The higher waste emissions in CAMS-REG-GHG v4.2 are indeed coming from the Silesia region (Fig. S7). There is no evidence of particularly large amounts of domestic waste or waste collection facilities in this area. The Silesia and Krakow regions report comparable amounts of municipal waste per

inhabitants, and in the same range as other regions of Poland (Statistics Poland, 2018). However, there is 5 times more waste from mining activities reported in Silesia than the other Polish regions (Statistics Poland, 2018). The emissions reported by CAMS are therefore associated with coal mining activities, especially mineral washing in the coal preparation plants. In our approach of distinguishing sources based on their isotopic signature, these emissions should be considered as fossil fuel related. However, in the CAMS inventory they are combined with waste emissions from the fermentation of organic substrate, which

have a distinctly depleted isotope signature (Table 2, Fig. 6.A). The emissions from on-site energy use for coal mining and for the manufacture of secondary and tertiary products from coal are included in the "other anthropogenic" category in both inventories (CRF sector 1.B.1.c, European Environment Agency (2019)). But in the EDGAR inventory, emissions categorised as from coal mining include fugitive emissions from the extraction and all the processing steps prior to combustion (CRF sector 1.B.1.a, European Environment Agency (2019)). They were therefore associated with the same signature as the coal extraction itself, which results in a better match with the observations than when using CAMS-REG-GHG v4.2.

itself, which results in a better match with the observations than when using CAMS-REG-GHG v4.2.

The isotopic signatures per peak obtained from the model are compared with the ones from the observations in Fig. 8. The histograms show the distribution of isotopic signatures from the Keeling plots applied to each peak we extracted from the measured and modelled time series. The correlation plots allow to compare the CH$_4$ peaks detected simultaneously in the observed and modelled time series.

When using the CAMS-REG-GHG v4.2 inventory, the $\delta^{13}$C source signatures varied between -52.4 and -48.5 ‰, a much more narrow range than from -55.3 to -39.4 ‰ for the observations. This reflects the over-representation of the waste category and its associated depleted $\delta^{13}$C signature. This bias towards depleted values is also visible in the $\delta^2$H signatures. The source signatures when using the EDGAR v5.0 inventory match the observations better: the average $\delta^{13}$C and $\delta^2$H of all elevations agree within their uncertainties, and the $\delta^{13}$C signatures are slightly correlated (r$^2$=0.33). The distribution of $\delta^{13}$C signatures

with EDGAR has a bimodal shape that we also observe in the measured data, but covers a smaller range of values. Some of the most enriched signatures in the observations are not reproduced by the model, for both $\delta^{13}$C and $\delta^2$H (Fig. 8). As shown in Fig. 6.A, $\delta^2$H allows to distinguish microbial fermentation from fossil fuel (or pyrogenic) sources, whereas the $\delta^{13}$C ranges for these 2 source types overlap. This suggests that the fossil fuel fugitive and combustion related emissions in the inventories are under-estimated. This corresponds to our findings from analysing the emission peak signatures of Fig. 7, and is consistent

with the lower $\chi$(CH$_4$) in the model compared to the observations described above (Fig. 5).

Finally, the absence of correlation between $\delta^2$H signatures from model and observations (Fig. 8.B) emphasises the need for more $\delta^2$H measurements in order to more precisely constrain the sources for this isotope signature. This limits the conclusions we could derive from measurements of $\delta^2$H.



## 4 Conclusions

This study presents measurements of $CH_4$ mole fractions, $\delta^{13}C$ and $\delta^2H$ of $CH_4$ in ambient air, performed continuously during 6 months in 2018 - 2019 at Krakow, Poland. The results were combined with model simulations from a high-resolution regional transport model based on two different emission inventories.

The source signatures of the pollution events observed in Krakow were compared with signatures from sources sampled around the study area. This allows us to identify the fossil fuel related sources as the main contributor to the $CH_4$ emissions.
The wind directions pointed towards Silesian coal mines, but the use of natural gas in the urban area of Krakow is also an important source. Our results showed that despite the presence of microbial $CH_4$ reservoirs, $CH_4$ of thermogenic origin contributes the most to the atmospheric emissions from the USCB mine shafts. Despite their variability, the $CH_4$ isotopic signatures of Silesian coal mines are generally well understood. This study significantly helps constraining the $CH_4$ isotopic signatures from the USCB coal mining activities. Our isotopic observations when the wind was from the west at relatively high
speeds confirm the prominence of coal related $CH_4$ emissions compared to biogenic ones (agriculture and waste).

In comparison to measurements made in the Netherlands (Röckmann et al. (2016), Menoud et al. (2020b)), the range of $CH_4$ isotopic signatures derived from the Krakow measurements was more enriched in $\delta^{13}C$ and $\delta^2H$, by 10 ‰ and 100 ‰, respectively. These large differences are directly related to the heterogeneity in the human activities impacting our climate: from agriculture (especially cattle farming) in the Netherlands, to the exploitation of fossil fuels in Poland. This provides additional
evidence for the value that the analysis of isotopologues can have in constraining the local to regional methane budget.

The $\chi(CH_4)$ computed using both inventories matched the measurements rather well ($r^2$=0.65 using EDGAR v5.0) during fall 2018. However, the agreement is less during the winter months ($r^2$=0.40), largely reflecting discrepancies in the timing of the pollution events. The model also under-estimated the $CH_4$ levels by on average 170 ppb compared to the observations. The isotopic results suggest that increased emissions in the inventories must be of fossil fuel origin.

The average isotopic source signatures from the model using the EDGAR v5.0 inventory were in good agreement with the ones from the measurements, which confirms the source attribution. Larger differences were observed on the level of individual peaks. Uncertainties remain because of the combination of different sources within one category in the EDGAR v5.0 inventory. Small discrepancies between observed and modelled signatures are also due to the inherent diversity of isotopic signatures, even within one source category, like we observed when sampling the USCB mines. But the emissions within the Krakow urban
area, where multiple $CH_4$ sources are detected at the study site, are affected in a particular way. The CAMS-REG-GHG v4.2 inventory quantified waste emissions as the main contributor to the regional $CH_4$ emissions, but does not distinguish residential waste from waste associated with the processing of coal, which resulted in a large bias towards isotopically depleted sources. Therefore, our method fails to assess in detail the performance of this inventory. Nevertheless we show the power of continuous isotope data for analysing $CH_4$ emission sources on monthly and daily scales, in a very detailed manner. The sensitivity of our
approach allows precise identification of the different sources. These measurements can be used in future work to improve and validate inventories, and help mitigation. This requires $CH_4$ sources to be characterised locally, and additional sampling campaigns in the city of Krakow would be required to better define the different sources and their isotopic composition.



Using $\delta^2$H measurements in the identification of the sources was more powerful in this region, compared to $\delta^{13}$C, as the $\delta^{13}$C from coal mine activities and the network gas overlaps with $CH_4$ emitted from microbial sources such as waste. Yet our
conclusions using $\delta^2$H isotopes are restricted by the limited amount of $\delta^2$H measurements available. Our $\delta^{13}$C data generally support the recent re-evaluations of global $\delta^{13}$C-$CH_4$ from fossil fuel sources towards less enriched values (Schwietzke et al. (2016)). The data presented here was collected in an area that has been under-investigated in the past, compared to its importance for the European $CH_4$ emissions. It is therefore an important contribution to studies on the global $CH_4$ budget. The high time resolution and temporal coverage of $\chi(CH_4)$, $\delta^{13}$C and $\delta^2$H in $CH_4$ provided by this data is also particularly helpful to
evaluate transport models on regional and global scales.

*Data availability.* The data that support the findings of this study are openly available at https://doi.org/10.5281/zenodo.4548748, upon request. It will be made open access once the article is published.

*Author contributions.* M.M., C.V. J.N. and J.B. performed the isotopic measurements. M.M., J.N. and M.S. performed the mobile surveys and sampling. M.M. and C.V. processed the experimental data. B.S. performed the model simulation. M.M. performed the analysis, drafted
the manuscript and designed the figures. T.R. aided in interpreting the results and worked on the manuscript. J.N., B.S., I.P., and T.R., discussed the results and commented on the manuscript. J.N., I.P., P.B. and T.R. were involved in planning and supervised the work.

*Competing interests.* No competing interests apply for this work.

*Acknowledgements.* This work was supported by the ITN project Methane goes Mobile – Measurements and Modelling (MEMO$^2$; https://h2020-memo2.eu/). This project has received funding from the European Union's Horizon 2020 research and innovation programme under the Marie
Sklodowska-Curie grant agreement No 722479. We specially thank all the team members of the experimental physics lab of AGH for their support in the installation and maintenance of the IRMS system.
We acknowledge ECCAD (Emissions of atmospheric Compounds and Compilation of Ancillary Data) for the archiving and distribution of the data.



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

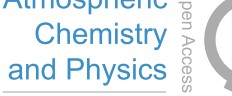

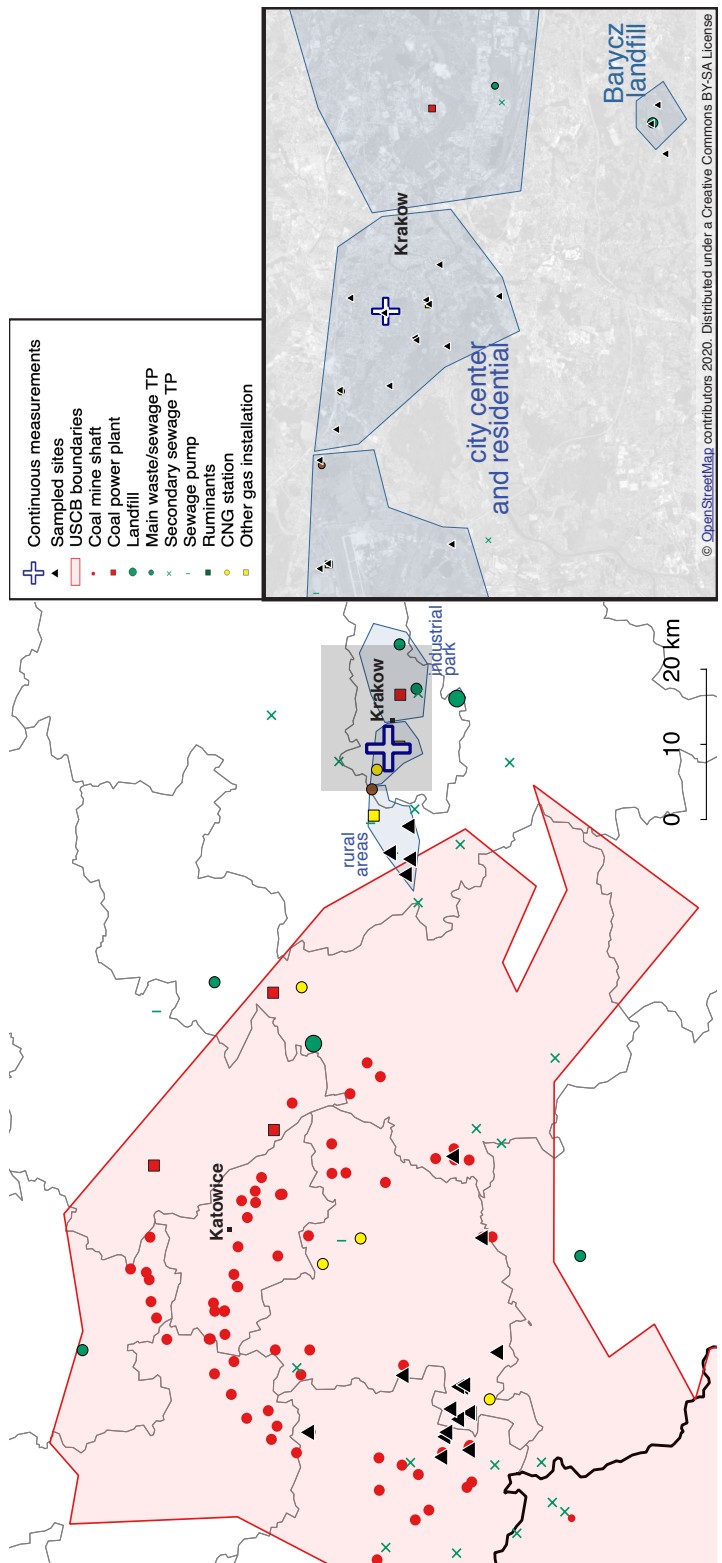

**Figure 1.** Location of the long-time measurements, sampled sites and potential anthropogenic methane sources. Note that this is not an exhaustive list: not all the sewage pumps are reported, and no official information on cattle farms was obtained. Other emissions from mining activities, coming from processing facilities or waste disposal, are not reported here. No $\chi(CH_4)$ enhancements were measured around stagnant water bodies, therefore they are not all reported here. ("TP" = treatment plant, "CNG" = compressed natural gas).





**Figure 2.** Time series of the observed $\chi(CH_4)$ (n=7886), $\delta^{13}C$ (n=3477), and $\delta^2H$ (n=4389), together with the $\chi(CH_4)$ time series observed at Kasprowy Wierch (green; n=21028). The shaded areas show when there was a mismatch between the IRMS and CRDS instruments in the mole fractions.



**Figure 3.** Monthly wind directions during the ambient air measurement period, at the same location. Bar lengths are percentages of records during the specified month (r-axis); colours define the $\chi(CH_4)$ range (legend).





**Figure 4.** Time series of the observed (blue circles) and modelled $\chi(CH_4)$, $\delta^{13}C$ and $\delta^2H$, based on the EDGAR v5.0 (red) and CAMS-REG-GHG v4.2 (green) inventories.

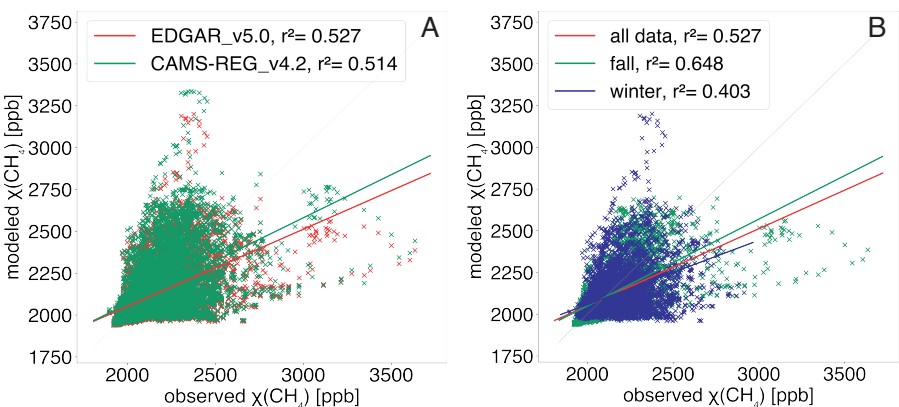

**Figure 5.** Correlation between observed and modelled $\chi(CH_4)$ values, using (a) the EDGAR v5.0 (red) or the CAMS-REG-GHG v4.2 (green) inventories, and (b) different time periods: fall (September 14 to November 15, 2018; green) or winter (November 15, 2018 to March 15, 2019; blue) computed using EDGAR v5.0.



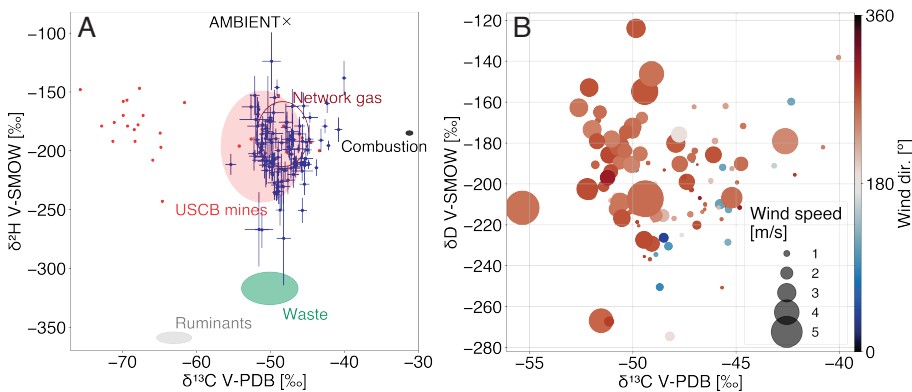

**Figure 6.** Dual isotope plots of the resulting source signatures from the $CH_4$ peaks identified in the time series. (a) Dark blue: source signatures with their associated $1\sigma$ uncertainties. Coloured areas: ranges of source signatures obtained from the collected samples. Red dots: source signatures of USCB coal gas derived from Kotarba (2001), Kotarba and Pluta (2009) and Kedzior et al. (2013). The combustion source signature is from coal waste burning samples reported in Menoud et al. (2020a). (b) Source signatures labeled by the average wind direction (colour) and speed (size) measured during the pollution event.

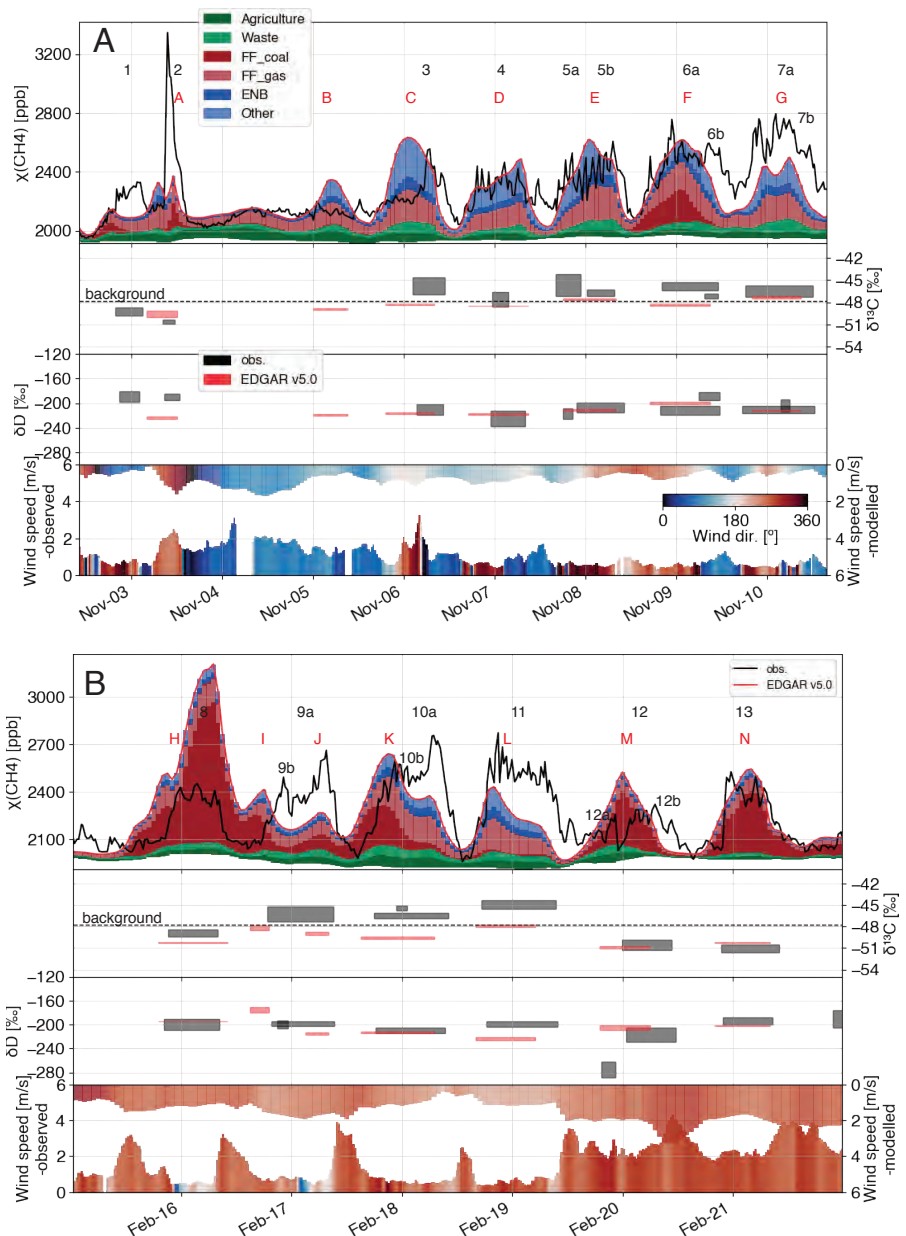

**Figure 7.** Detailed analysis of two subsets of the dataset, (a) from Nov. 2 to 10, 2018, (b) from Feb. 15 to 22, 2019. Top panels: observed (grey) and modelled (red) mole fractions and relative source contributions from the EDGAR v5.0 inventory. Middle panels: $\delta^{13}$C and $\delta^{2}$H source signatures of individual peaks of the observed (grey, from peak 1 to 13) and modelled (red, from peak A to N) time series. Box heights represent $\pm 1\sigma$ of each peak isotopic signature. Bottom panels: wind speed and direction measured simultaneously at the study site (pointing up), and used for the CHIMERE simulations (pointing down).

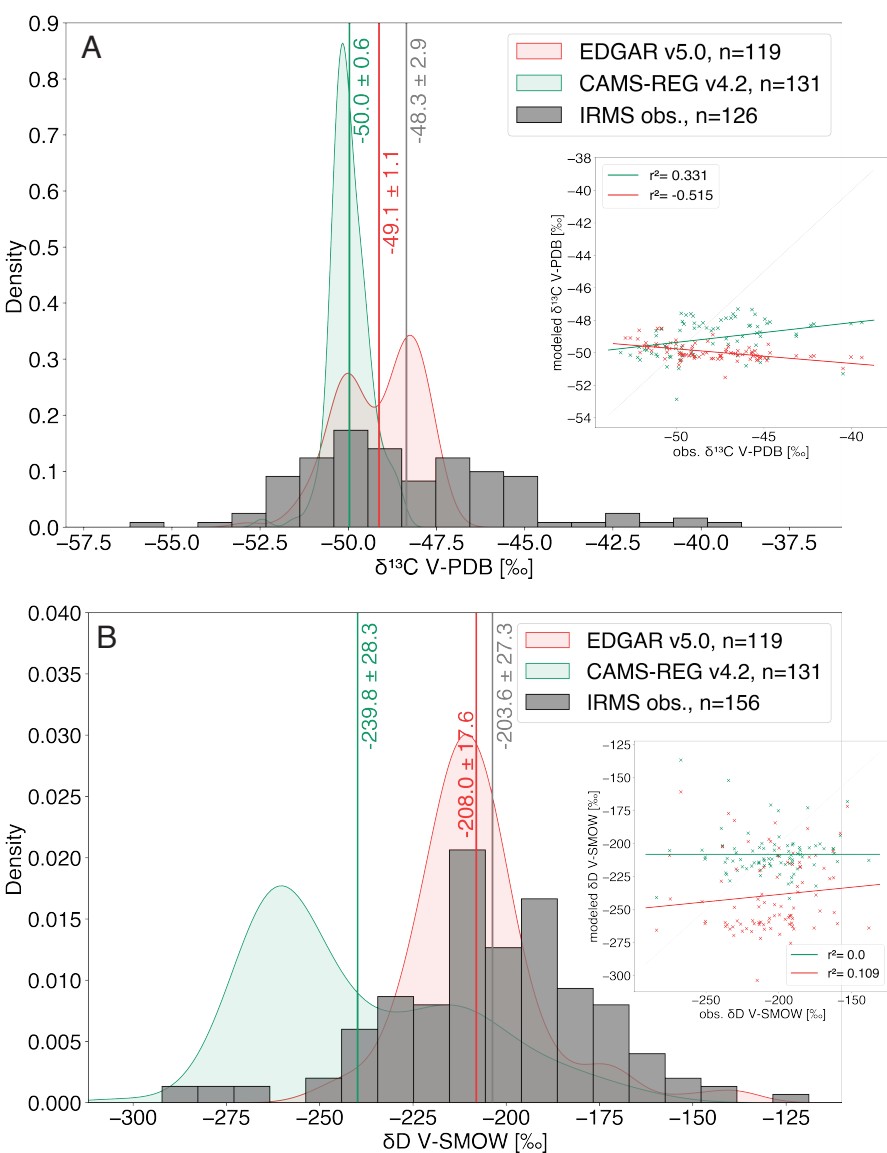

**Figure 8.** Distribution of source signatures of all peaks, and in the inset the correlation between modelled and observed ones. The vertical lines show the average values of each distribution ($\pm$ $1\sigma$). (a) $\delta^{13}$C signatures in the observed (grey, n=126), modelled using EDGAR v5.0 (red, n=119) and modelled using CAMS-REG-GHG v4.2 (green, n=131) time series. (b) $\delta^{2}$H signatures in the observed (grey, n=157), modelled using EDGAR v5.0 (red, n=119) and modelled using CAMS-REG-GHG v4.2 (green, n=131) time series.





**Table 1.** Methane emission categories considered for this study, with the corresponding classification in the inventories, and the respective isotopic signature used to compute $\delta^{13}$C and $\delta^2$H time series with CHIMERE. If no references are specified, the assigned isotope values are derived from the sampling campaigns we carried out in the study area as described below.

| CHIMERE source category | CRF sector[1] | IPCC 2006 code | EDGAR v5.0 sector | CAMS-REG-GHG v4.2 sector | Assigned $\delta^{13}$C V-PDB [‰] | Assigned $\delta^2$H V-SMOW [‰] |
|---|---|---|---|---|---|---|
| Agriculture | 3 Agriculture | 3A1 / 3A2 / 3C1b / 3C2, 3C3, 3C4, 3C7 | Enteric fermentation / Manure management / Agriculture waste burning / Agriculture soils | K Agriculture-livestock / L Agriculture-other | -63 | -359 |
| Waste | 5 Waste | 4A, 4B / 4C / 4D | Solid waste landfills / Solid waste incineration / Waste water handling | J Waste | -51.6 | -299 |
| Fossil fuels | 1B Energy - Fugitive emissions from fuels | 1B1a / 1B2bi, 1B2bii / 1B2aiii2, 1B2aiii3 | Fuel exploitation, coal / Fuel exploitation, gas / Fuel exploitation, oil | D Fugitives | -51 / -48.5 / -49.3 | -192 / -194 / -193 |
| Non-industrial combustion | 1A4, 1A5 Energy - Other sectors, Other[2] | 1A4, 1A5 | Energy for buildings | C Other stationary combustion | -32.1[3] | -185[3] |
| Other anthropogenic | 1A Energy - Industries | 1A1a / 1A1b, 1A1ci, 1A1cii, 1A5biii, 1B1b, 1B2aiii6, 1B2biii3, 1B1c / 1A2 | Power industry / Oil refineries and transformation industry / Combustion for manufacturing | A Public power | -49.3 | -193 |
|  | 2 Industrial processes and product use | 5B / 2B / 2C1, 2C2 / 2D3, 2E, 2F, 2G | Fossil fuel fires / Chemical processes / Iron and steel production / Solvents and products use | B Industry / E Solvents | | |
|  | 1A3 Energy - Transport | 1A3b / 1A3d / 1A3a / 1A3c, 1A3e | Road transportation / Shipping / Aviation / Railways, pipelines, off-road transport | F Road transport / G Shipping / H Aviation / I Off-road | | |
| Wetlands | | | | | -73.2[3] | -323[3] |
| Background | | | | | -47.8 | -89 |

[1] European Environment Agency (2019)
[2] Mostly the use of coal for heating households (European Environment Agency (2019))
[3] Menoud et al. (2020ab)





**Table 2.** Isotope signatures of the different sources sampled in the region surrounding the study site.

| Source type | Number of sites | Mean $\delta^{13}$C V-PDB [‰] | $1\sigma$ | Mean $\delta^2$H V-SMOW [‰] | $1\sigma$ |
|---|---|---|---|---|---|
| Coal mine | 16 | -51.0 | 7.1 | -191.6 | 27.8 |
| Cow barn | 1 | -63.0 | | -358.7 | |
| Landfill | 2 | -55.4 | 0.8 | -275.0 | 34.5 |
| Manhole | 8 (5/3) | -45.0 (-42.5/-49.1) | 9.0 (10.9/3.1) | -233.7 (-176.4/-329.2) | 81.0 (21.1/12.3) |
| Network gas | 7 (1) | -48.5 (-51.4) | 2.9 (0.4) | -193.6 (-205.0) | 17.3 (0.001) |
| Unknown | 23 | -49.0 | 6.2 | -195.3 | 39.8 |





**Table 3.** Methane absolute emissions and contributions of the different source categories used in CHIMERE to the total simulated $\chi(CH_4)$, for the EDGAR v5.0 and CAMS-REG-GHG v4.2 inventories.

| | Emissions over domain [TgCH$_4$/yr] | | Contribution at Krakow [ppb/ppb] | |
|---|---|---|---|---|
| Source categories | EDGAR v5.0 | CAMS-REG-GHG v4.2 | EDGAR v5.0 | CAMS-REG-GHG v4.2 |
| Agriculture | 2.02 | 1.64 | 0.168 | 0.114 |
| Waste | 1.88 | 1.22 | 0.142 | 0.438 |
| Fossil fuels - coal | 0.52 | - | 0.145 | |
| Fossil fuels - gas | 1.23 | - | 0.309 | |
| Fossil fuels - oil | 0.02 | - | 0.00226 | |
| Fossil fuels - total | 1.77 | 1.32 | 0.456 | 0.346 |
| Non-industrial combustion/Energy for buildings | 0.31 | 0.28 | 0.0986 | 0.0667 |
| Other anthropogenic | 0.09 | 0.16 | 0.118 | 0.0201 |
| Wetlands | 0.4 | | 0.0178 | 0.0157 |
| Total | 6.07 | 4.64 | 1 | 1 |