# Peer review of "Methane (CH4) sources in Krakow, Poland: insights from isotope analysis"

_Atmospheric Chemistry and Physics, 2021_

## Author Response (AR1)

**Review of Menoud et al: Methane (CH4) sources in Krakow, Poland: insights from isotope analysis**

\*\*\*\*\*\*\*\*\*\*\*\*\*\*\*\*\*\*\*\*\*\*\*\*\*\*\*\*\*\*\*\*\*\*\*\*\*\*\*\*\*\*\*\*\*\*\*\*\*\*\*\*\*\*\*\*\*\*\*\*\*\*\*\*\*\*

**RC1**

\*\*\*\*\*\*\*\*\*\*\*\*\*\*\*\*\*\*\*\*\*\*\*\*\*\*\*\*\*\*\*\*\*\*\*\*\*\*\*\*\*\*\*\*\*\*\*\*\*\*\*\*\*\*\*\*\*\*\*\*\*\*\*\*\*\*

**General comments**

This is a nice paper with detailed analysis of methane and d13C-CH4 and d2H-CH4 from a region with under-constrained methane sources. I think that there are many edits that could made to make the study easier to follow for the reader – it gets a bit confusing as to which measurements are from what kind of sampling, and in some places, what is modeled and what is measured.

I would have appreciated a lot more detail as to measurement uncertainty, and I am a bit troubled by the 70 ppm offset in the two instruments measuring methane. Also, Figure S2 was really confusing. It seems like some kind of quality control could have prevented a situation where you're not sure which instrument is making the accurate measurement and which one has the problem. Some other metrics of data quality (ie, accuracy and precison of a known unknown) would have been helpful for interpreting the other data presented here. Furthermore, I found Tables 1 and 2 a bit confusing – it was hard to tell what was measured in this study versus others, how many measurements were included, how they correspond to figure 6A. When there were few measurements (ie, wetlands), maybe other studies should have been incorporated. Also, sources were measured with just a few samples when the mobile sampling was above ambient; but the isotopic signature still consists of a contribution from background. Some acknowledgment of this (or a better, Keeling-type strategy) seems necessary.

I also would have liked to see more use of the Miller-Tans plots – perhaps using wind direction or seasons as a method to separate source signatures.

Although Figure 7 is really nicely plotted, with a ton of information, I found that the text seemed to sound like a lot of hand-waving. Some editing will help with clarity (ie., it was not always clear when you were referring to Feb or to Nov). Also, the distribution of sources in the model doesn't seem to address uncertainty in the source attributions – some acknowledgment of this might have tempered the conclusions. However, I think that acknowledgment of the uncertainties doesn't take away the gist of the conclusions, ie local vs USCB sources.

In sum, this is a really interesting, detailed look at methane sources in Krakow, which highlights both the utility of methane stable isotopes the limitations, and the need to make more measurements at regional scales. I think it's important not to overstate conclusions (note my word choice suggestions in the conclusions), and to acknowledge the difficulty in separating isotopically-overlapping sources (ie., coal and natural gas). And yet the isotopes clearly show where the inventories are wrong, as well as specific mis-attributions of sources, and this is very useful. I appreciated the comparison to the Netherlands studies, where different methane sources are dominant. Further discussion of how approaches like these could be used in mitigation efforts would be really interesting.

Thanks for the opportunity to dive in to this work.

AC: Thank you for your comments and relevant suggestions.
We've answered the general issue on the clarity of what was done by including the details on the Keeling plots from which we derived the source signatures of the locations sampled during the mobile surveys. We've added the criteria on selecting the locations with significant source signatures. The Miller-Tans plots were applied on different range of wind directions, which deepened our analysis. The conclusions were revised and tempered.

**Specific comments**

P1 L18: you don't name the second inventory in the abstract – so, Edgar is better than what?
AC: Yes this can indeed be confusing. We added the name of the second inventory.

P2 L24: "Greenhouse gas" is somewhat colloquial – "climate forcing trace gas" is more meaningful. If you want to use GHG, perhaps define it better
AC: Greenhouse gas is a scientific term that is used a lot, in scientific publications but also in the IPCC report. We think it is less complicated to use this term, and added its definition for more clarity.

P2 L25: "negative consequences" is a bit vague
AC: This was changed to "negative consequences of climate change on people and societies".

P2L52: Can you summarize the previous studies from USCB?
AC: Yes, we added some sentences describing the main outcomes of these studies. In the conclusion, we now compare our isotopic signatures for the USCB with values defined by Galkowski et al. 2020.

P3L65: The USCB is pretty big – is the center 50 km west of Krakow, or the eastern edge? This could be more descriptive. Also, Figure 1 is not very high quality – I can't read the names of the towns, so maybe leave them off?
AC: This has been adjusted. We have changed the background on the map in Figure 1 and made sure all the text is readable.

P3: There is no description here of Kasprowy Wierch, so when I saw the results I was confused
AC: The measurement site of Kasprowy Wierch is now introduced in the method section.

P3L83: One to three samples … clearly not enough for a Keeling plot, so were these assumed to be entirely source methane?
AC: In combination with one or two background samples, the number of points per Keeling plots were from 2 to 5. We are confident on the obtained source signatures for several reasons:
- only 5 of the Keeling plots were made with 2 points. These concerned locations where the $x(CH_4)$ levels of the source were very large: from 1000 ppm to 99% (pure gas). Therefore, the source signature value almost corresponds to the measured value of the sample.
- for the scientific analysis we only kept the plots where the points aligned well, and analysed the standard deviation of the derived intercept, the Pearsson coefficient r2, and the maximal mole fraction relative to background. These criteria are now detailed in the methods.

Figure S2: Is there a pump that pulls the sample through? Are the arrow shapes vents? Valves?
AC: We assumed that this comment concerned Figure S1. The pumps are now indicated.

P4L99: I'm curious about the calibration of the IRMS mole fraction measurements (especially since it becomes an issue) – can you expand on this? Is it a one-point calibration? Can you point to the accuracy/precision of this method? Also, can you cite accuracy/precision on the Picarro? Which should I trust more?
Also, nothing about the accuracy/precision about the isotopic measurements?
AC: We've added value of the reproducibility of our measurements, that we obtained from measuring air samples multiple times. The precision of the Picarro as reported by the

manufacturer was also added. We hope this improves the description of our system.

Table 1: I see a reference 2 but no data pointing to it.
AC: The reference 2 points at the "Energy - Other sectors, Other", in the "CRF sector" column.

P6L152: Considering that you have different wind directions for different times of the year, is it appropriate to use M-T plots over the entire dataset?
AC: The Miller-Tans plot was used here to reflect the average source signature over our time period, regardless of the wind direction. It is true that since the wind was mostly from the west, the resulting signature is more representative of emission sources to the west than from any other direction. We made sure this is clearly explained throughout the paper. We've also computed one regression per wind direction category, as you suggested. We've updated the text and the figure in the supplementary material accordingly.

P6L168: Yikes, this is a huge mismatch. What is the mismatch at other times? You corrected the IRMS data? Also, figure S2 is really confusing – I don't understand the x axis. Was there any qc through either instrument (a surveillance cylinder?) that would have caught a problem in either instrument?
AC: At other times, the mismatch was of -6.4 ± 16 ppb between IRMS and CRDS. It was statistically not significant, and could only be due to the hourly averaging we need to apply in order to make the data comparable.
We tried different ways to correct the IRMS data at the periods of the mismatch. For example by applying a constant offset, an offset calculated per day, or a linear function. The resulting source signatures were not affected significantly, so we decided to keep the original data. We added the results of the source signatures obtained when applying the offset in Figure S5. We hope it is now easier to see that it compares well with the original data, and removed Figure S2.
We didn't use a surveillance cylinder, but the Picarro was calibrated regularly because it was also used in the field. It was not continuously connected to the ambient air line. We don't doubt about the instrument precision, and we explored different potential causes of the mismatch and couldn't find what happened. It is not ideal, but we can show this doesn't affect our analysis of the IRMS results and this study's conclusions.

P7L200: Any signs of seasonality in the background? (maybe it was hard to see in 6 months, but seasonal cycles are quite evident that far north)
AC: There was no clear seasonality in the background. The background values calculated per month were sept.: 1951 ppb, oct.: 1954 ppb, dec.: 2000 ppb, jan.: 1978 ppb, feb.: 2038 ppb, mar.: 1963 ppb.

P8L216. This sentence implies that pollution events and diurnal cycle are related. Are they? Maybe you should show the lack of diurnal cycle in winter in figure S3?
AC: We have changed the formulation to make it clearer: there was less regularity in the observed CH4 enhancements in winter. It is indeed because of a weaker diurnal cycle, that is now illustrated in figS3.

P8L218: If you are referring to the slopes, maybe print them on the plots with the r2?
AC: We do not interpret the absolute values of the slope, just whether the line is below or above the 1:1 line. This is visible on the figure without adding more numbers.

Figure 6A: Blue is the keeling plot derived source signatures, correct? Can you explain where the uncertainties came from? Why don't the red or black dots have uncertainties? Colored ovals: these are from the bags you collected on your mobile campaign? (But not a Keeling plot analysis, just the samples themselves?) The data from the ovals are in table 2?

How did you decide on the shape of the wetlands oval, since you have just one point? Can you use another study?

Speaking of table 1, assigned d13C and d 2H, it would be helpful if you listed the number of samples and associated error (or point the reader to table 2 …)

AC: The uncertainties are the standard deviations of each Keeling plot intercept, as explained P6L158. The black ellipse shows the pyrogenic CH4 source signature reported in Menoud et al. (2020) database we used for this study. So as for the ruminants, it is the value used for this emission category, not only for one location so I used the oval shape, with a size of 1 order of magnitude larger than the precision of our isotopic measurements. The red dots are values from the literature, and are not always reported with uncertainties. Also, their purpose in the figure is for comparison with our results only.

Table 2: Some of the data in table 2 are also in table 1… so this is confusing.

AC: The data in Table 2 was used to determine the input values for the model, which are reported in Table 1. We think it is important for the reader to see the link between them and some explanations were added to make it less confusing.

P8L238: Maybe refer to mobile sampling as you did in the methods

AC: This has been added

P8L247: Agreed to within 10 % or 10 permil. This is confusing.

AC: The differences are now always reported in ‰ (per mill).

P9L265: Would ethane or another measurement help to confirm this?

AC: Ethane might help with this distinction. The isotopically depleted natural gas indicates a microbial origin, so the ethane content is expected to be low.

P9L266: I'm not sure where to find these data

AC: References to the appropriate table and figure have been added.

P9L274: You sampled outside the shafts if the methane was 200 ppb above background … so your "signature" still contained a lot of background. I don't think you've reported how far above background these plumes were.

AC: More details on the CH4 range we sampled are now added at the beginning of the paragraph.

P10L294: How do you determine the percentage above background?

AC: Out of all the peaks we identified, 40.5% of their isotopic source signatures were higher than the d13C value of the background (-47.8 ‰) (P7L198).

Figure S6: So each data point plotted here that is not an X (literature value) is one bag of collected sample?

AC: Each point that is not an X represents a sampled location, but not 1 single sample. Several samples + 1 or 2 background are needed to derive each source signature on the graph.

P10L313: "In general" maybe should be "In November"? I'm confused

AC: Not necessarily. Although there was a period with only eastern winds in November, eastern winds also occurred at other times.

P10L316: "The modelled peaks C,D,E, and G". Also, why no discussion of H, it's a big one?

AC: The peaks C,D,E and G have in common "a large contribution from natural gas and from the "other anthropogenic" categories". Peak H is discussed later in the paragraph (P11L351) together with other peaks with a large contribution of coal related sources.

Figure 7: What is ENB?
AC: The acronym ENB was indeed not clear. The category was changed to correspond to the name given in table 1.

P12L382: "the modelled isotopic signatures with CAMS are always more isotopically depleted in d13C"… but wait, waste emissions have a large influence? Or just in the CAMS inventory? It is unclear what is inventory/model and what is real.
AC: The magnitude of waste emissions is larger in the CAMS inventory. This is because of a different classification of emissions related to the processing of coal. The sentence was re-formulated to make the distinction between model and inventory clearer.

P13L409: R2 describes the variance; to say that something is correlated you need to look at the slope of the line and whether it's statistically different than 0.
AC: We think R2 actually is a tool to evaluate if something is correlated. From WIKIPEDIAC: the coefficient of determination, denoted R2 or r2 and pronounced "R squared", is the proportion of the variance in the dependent variable that is predictable from the independent variable(s).

P13L416: It's interesting that you need hydrogen isotopes to separate fossil fuel from biogenic sources, yet you say that you are limited by what you can do with d 2H.
AC: Yes, this is not incompatible. d2H appeared to be very useful, despite the large uncertainties. So if we lower the uncertainties, we might be able to separate even more sources.

P14L428: Well-understood … maybe. Your measurements from the USCB mines don't agree very well with previous studies.
AC: "understood" was not the right word indeed. The processes are not understood in detail, but the variability is well constrained and the overall emissions from the USCB are well characterised. We've changed the text accordingly.

P14L436: I'm not sure about "rather" well. Maybe somewhat?
AC: This was changed to "relatively".

P14L450: Again, "precise identification" seems like a bit of a stretch.
AC: This was adjusted accordingly.

P14L451: "help mitigation" seems a bit vague – can you expand on this? It seems like identifying leaks is an obvious way that your method could be useful
AC: "help mitigation" would here be possible through the improvement of inventories. Identifying leaks is another way, somewhat related. We've clarified this in the text.

P15L453: d 2H more powerful than d13C in this study? But then you say that d 2H conclusions are limiting. So maybe you want to say that it was essential to have d 2H measurements in this case, without saying that they are "better" than d13C.
AC: Thank you for this suggestion. The text was changed accordingly.

Technical corrections
In general: please be careful about the use of "depleted" and "enriched" without specifying which isotope you are referring to.
P2 L 29: "of the ones of CO2" is awkward
AC: This was adjusted accordingly.

P2 L 30: "total CH4" not "the total CH4"

AC: This was adjusted accordingly.

P2 L39: "scales" can go before the references
AC: This was adjusted accordingly.

P2 L54: Can you merge the sentences "In this study we investigate …. " and "we attempted"; because they are the same effort, not two different goals, correct?
AC: These goals are quite distinct: on the one hand we want to isotopically distinguish the sources (no spatial component), and on the other hand we want to see what CH4 emissions we detect in Krakow, and the link with Silesia (particular geographical context). We think it is less confusing to keep 2 separate sentences here.

P3 L60: compare not compared
AC: This was adjusted accordingly.

P5L120: methane mole fractions
AC: This was adjusted accordingly.

P5L124: "enhancements" instead of elevations?
AC: This was replaced throughout the text.

Figure S1 label: steel not steal
AC: This was adjusted accordingly.

P7L191: "mainly from the west, with a small contribution (x %?) from the east/northeast"
AC: This was adjusted accordingly.

P7L211: the fall of 2018 (or fall 2018)
AC: This was adjusted accordingly.

Figure 2: your legend is 'IRMS' and 'Kasprowy Wierch'. A legend of 'IRMS, Krakow' and 'Picarro, Kasprowy Wierch' might be better.
AC: This was adjusted accordingly.

Figure S4: no comma after "measurement period". "Bar lengths are percentages of records from that wind direction during the specified month"
AC: This was adjusted accordingly.

Figure 3: I think I see a few differences in the CH4 mole fraction data between this plot and Figure 2 early in the record. Probably doesn't matter, but presumably you intend to show the same record?
AC: This has been verified.

P8L224: too-low (or insufficient)
AC: This was adjusted accordingly.

P8L220: "When using CAMS in the model, …". This paragraph could be more clear in general.
AC: This paragraph was rephrased.

P8L234: Don't start a sentence with a number.
AC: This was corrected.

P9L251: Proportions vary or proportion varies

AC: This was corrected.

P9L252: too many commas – awkward
AC: The sentence was split in 2 sentences.

Figure S6 legend: "which peak isotopic signatures did not significantly differ"
AC: This was rephrased.

P9L270: delete "found"
AC: This was changed accordingly.

P9L280: in regard to (instead of "as regards")
AC: This was changed accordingly.

P10L306: add "at Lutjewad"
AC: This was added.

P10L312" extra spaces
AC: This was changed accordingly.

P10L316: "The modelled peaks C,D,E, and G"
AC: This was changed accordingly.

P11L346: "depleted of heavy isotopes
AC: This was changed accordingly.

P13L390: There are 5 times
AC: This was changed accordingly.

P13L405: don't need a new paragraph.
AC: This was changed accordingly.

P13L413: combustion-related
AC: This was changed accordingly.

P13L414: findings from analyzing the emission peak signatures" =wordy and unclear
AC: This was rephrased.

P14L424: fossil fuel-related
AC: This was added.

P14L430: coal-related
AC: This was added.

P15L457: were collected
AC: This was changed accordingly.

P15L459 : are particularly helpful
AC: This was changed accordingly.

**RC2**
* * *
**General comments**

New measurements of methane and its two most abundant stable isotopes, 13CH4 and CH3D, are presented from a site in Krakaw, Poland. Some new measurements of source isotopic signatures are also carried out. The measurements are analysed, using a chemical transport model. It was found that the observations are strongly influenced by a mixture of sources from the nearby urban area, and fossil fuel extraction emissions from the nearby Upper Silesian Coal Basin.

The paper is well written, and the analysis of the data is thorough and sound, as far as I can see. The main weaknesses of the paper are that only a relatively short measurement record was obtained, and the site, being situated in a densely populated urban area, doesn't appear to be ideally suited to understanding regional emissions (e.g., I presume the relatively poor fit of the data and model are due, at least in part, to the proximity of unresolved sources in the model). However, the authors have done a good job of extracting as much information as possible from this dataset. I recommend the paper for publication in ACP, following some relatively minor changes.

AC: Thank you for your comments. In the conclusion, we've emphasised the difficulty of distinguishing the different sources in the urban context. This is indeed a limitation, but the analysis of the wind data still allows to conclude on the USCB influence, as our site was west of Krakow, just downwind of Silesia.

**Specific comments**

L1 – 2. I think the first two lines could be cut as they are quite general, and there would be too much to unpack in the assertion that methane emissions are a threat to the adherence to the Paris Agreement "goals".

AC: We think it is still useful to remind the reader of the motivation behind studying the methane budget. The importance of reducing CH4 emissions in order to follow the Paris agreement goals was not only mentioned in several other studies, but also demonstrated in Nisbet et al. 2019. We've added the reference to this paper. Because it is indeed not our main point here, we moved the reference to the Paris Agreement goals from the abstract to the introduction.

L16: "The X(CH4) are generally under-estimated in the model". This should be more specific. Do you mean the magnitude of the pollution events are under-estimated? (I assume so, as this is the only part that's relevant to the regional emissions covered here. We don't really care if the model gets the background component correct).

AC: Indeed, it is the magnitude of the pollution events that is under-estimated, and it is what we wanted to emphasised. We've changed the text accordingly.

AC: There are several potential causes of the over-estimation of CH4 mole fractions in the model, and they are discussed P8L216-227. It is indeed most likely that the magnitude of emissions is too low in the inventory, but it is not something we investigated, quantified and verified in this study. We still specified this statement as you suggested.

L17: "… would lead to better agreement". Need to say what the better agreement is with respect to (I.e. the data).

AC: This was changed accordingly.

L29: "only 3% THOSE OF CO2…" (instead of "of the ones of")

AC: This was changed accordingly.

L65: I suggest "consists of", rather than "gathers"

AC: This was changed accordingly.

L93-94: "…, as described in…" (delete "the one")
AC: This was changed accordingly.

L180: "This is due to a lowering of the boundary layer when the temperature decreases in the evening". This isn't technically correct. It's the temperature gradient in the lower atmosphere that leads to a lowering of the boundary layer height, rather than the temperature itself. I.e. the atmosphere tends to become more stable at night time.
AC: This was rephrased.

L182: "emission peaks". Need to be careful with terminology here. You aren't describing "emission" peaks, but "concentration"/"mole fraction" peaks.
AC: This was changed accordingly.

L216: I don't think the authors mean that pollution events are "less predictable" here. This implies that meteorological forecasting is less skilful in winter, which I don't think is the point they are trying to make.
AC: This was reformulated.

L240 and elsewhere: I'm not actually sure what "manholes" means in this context. Perhaps this is a technical term, but, to me, a manhole is one of the many holes you see in the street that give access to the sewer system, etc. Is there a more descriptive term that can be used? If not, a line clarifying what this means would be helpful.
AC: We don't come up with any better term, but what the word "manhole" refers to was now introduced (Table 2). We don't want to use a more descriptive term because based on our isotopic measurements, the emissions from these holes were not clearly associated with one type of source.

L347: "westerly", rather than "western" winds.
AC: This was changed accordingly.

L434: I think you need to be clearer that this statement comparing the source mixture in the Netherlands and Poland only applies to these two parts of the two countries. I.e., you can't be sure (as far as I'm aware) that there are regions of the Netherlands that are more strongly influenced by fossil fuels, or vice versa.
AC: This was adjusted accordingly.

L440: "which confirms the source attribution". It's not clear from this sentences what is being confirmed.
AC: This was reformulated.

L444: "But the emissions within the Krakow urban area, where multiple CH4 sources are detected at the study site, are affected in a particular way." I don't know what this sentence means. In what way are they affected?
AC: The sentence was replaced by: "When multiple CH4 sources contribute to the total x(CH4), as it was the case for the Krakow urban area, the uncertainties in the isotopic characterisation increase further."

**List of relevant changes**
* * *
**Manuscript**
- P6: details on the regression methods were added, and how the Keeling plots of the sampled locations were performed. For consistency, all Keeling plot regressions were now only made with the Orthogonal Distance Regression method, which resulted in slightly different values throughout the manuscript without changing the interpretation.
- Table 1: the input values changed because they are derived from the sampled source signatures, as in Table 2.
- Figure 6, 7 and 8 appear different due to the updated source signatures form the distance regression method.
- P14: the conclusion from the detailed analysis of both measurement and model data have impact on our interpretation of emission inventories. This was further emphasized to justify the classification of the manuscript as research article and not measurement report.

**Supplement**
- Figure S2 removed, Figure S5B contains the same information in a clearer way.
- Figure S4 includes separate regressions depending on the wind directions, as well as the resulting signatures.
- Table S1 was added to bring more information on the isotopic source characterisation from the sampling.

---

## Referee Report (RR1)

**General response to revisions:** I think that M. Menoud and co-authors have done a thorough job responding to my suggestions and edits. I appreciate the additions to the methods, which help clarify the results later on, and provide relevant analytical uncertainty. The extra keeling plot table is helpful, as is the wind direction in the Miller-Tans analysis (though see comment below about the axes labels). I think that the verbiage in discussing isotopes needs a bit more cleaning up (see "enriched" and "depleted" comments below), but overall the edits have made for a paper that is easier to understand and follow. I have a few more suggestions for edits below, but this is almost ready for publication and will be well received.

**Text:**

L6: "to the east"?

L25: gases not gasses, here and throughout.

See https://writingexplained.org/gases-or-gasses-difference

L111: "filled with air with …" is a bit awkward

L113: delete the "of" in "measurements is of 16 ppb"

L171: see comment about the equation for miller-tans plots in the figures section

L180: there is a redundant phrase in here

L182: standard deviation *was* lower

L262: "Depleted and enriched: I wasn't clear enough in my suggestion as to the use of "Depleted" and 'enriched". You're not supposed to use the words without defining the isotope specifically (see Coplen 2011 reference below). So, instead of saying "microbial sources are more depleted than thermogenic ones" you need to say "microbial sources are more depleted in $^{13}C$ than thermogenic ones". If it is clear that you're talking about carbon isotopes, you could say "microbial sources are more depleted in the heavy isotope than thermogenic ones". With all of your discussions, it might start to feel wordy, in which case you can talk about delta values being more negative or positive. Please fix these throughout your text. For instance,

L356: not only *do*

L373: "shifted toward values more depleted in heavy isotopes" or "shifted toward more negative values"

Coplen, T. B. (2011). Guidelines and recommended terms for expression of stable-isotope-ratio and gas-ratio measurement results. *Rapid communications in mass spectrometry*, *25*(17), 2538-2560.

L315: Of the CH4 peaks, 40.5 % were more enriched in $^{13}$C than the background values of -47.8 ‰.

L451: "The wind directions pointed toward Silesian mines" might confuse your readers ..

L457: Is dD=-225 Galkowski's value? Please clarify

L457: "helps constrain"

**Figures:**

Figure S2: Legend should say fall and winter.

Figure S5: I like this analysis. But, I just want to make sure I understand you correctly – do you subtract a background? As I understand it, the equation for MT plots is

$(d_{obs}*C_{obs})-(d_{bg}*C_{bg})=d_s(C_{obs}-C_{bg})$

(where d=delta)

I assume that you are doing this right, but if so, your Y axis would be $(d_{obs}*C_{obs})-(d_{bg}*C_{bg})$ and your x axis would be $(C_{obs}-C_{bg})$.

Figure S6. I like the improvements – seems more clear. I don't see a label for the triangle – are you still plotting by date?

Table S1: Thanks for showing the original data. Why do you have negative r^2 values? Also, there is a typo in the title: "This happens when the sampled source and background δ13C values *are* very close, so that the slope of the fit line is close to 0."